# Unique Case Report: A Rare Association of 21-Hydroxylase Deficiency with Triple X Karyotype

**DOI:** 10.3390/genes16030354

**Published:** 2025-03-20

**Authors:** Rossana Santiago de Sousa Azulay, Alexandre Nogueira Facundo, Sarah Sousa e Sousa, Gilvan Cortes Nascimento, Marcelo Magalhães, Clariano Pires de Oliveira Neto, Joana D’arc Matos França de Abreu, Débora Cristina Ferreira Lago, Sabrina da Silva Pereira Damianse, Viviane Chaves de Carvalho, Caio Andrade Nascimento, Vandilson Pinheiro Rodrigues, Fernanda Borchers Coeli-Lacchini, Margaret de Castro, Manuel dos Santos Faria

**Affiliations:** 1Service of Endocrinology, University Hospital of the Federal University of Maranhão (HUUFMA/EBSERH), São Luis 65020-070, Brazil; rossana.azulay@ufma.br (R.S.d.S.A.);; 2Research Group in Endocrinology and Clinical and Molecular Metabolism (ENDOCLIM), São Luis 65020-040, Brazilvandilson.rodrigues@ufma.br (V.P.R.); 3Department of Medicine, Ribeirao Preto Medical School, University of Sao Paulo, Sao Paulo 14048-900, Brazil; fbcoeli@gmail.com (F.B.C.-L.); castrom@fmrp.usp.br (M.d.C.)

**Keywords:** congenital adrenal hyperplasia, 21-hydroxylase deficiency, triple x karyotype, primary ovarian failure

## Abstract

**Background**: Congenital adrenal hyperplasia (CAH) represents a group of autosomal recessive disorders characterized by impaired cortisol synthesis in the adrenal glands. Over 90% of CAH cases result from a deficiency of the enzyme 21-hydroxylase (21OHD). The clinical spectrum of 21OHD ranges from the severe, life-threatening salt-wasting classic form, often presenting with prenatal virilization in females, to the non-classic (milder) form, which lacks glucocorticoid deficiency. Females with the non-classic form may experience symptoms of hyperandrogenism or infertility later in life, while males with non-classic CAH are often undiagnosed due to the subtler presentation. The coexistence of genetic anomalies and CAH is rarely reported in the literature, particularly in cases involving Triple X syndrome—a condition typically associated with a mild and frequently underdiagnosed clinical course. **Case presentation**: Here, we present a unique case of a 38-year-old woman with a history of premature ovarian failure and subsequent clinical features of hyperandrogenism. Further investigation revealed a novel association between partial 21OHD and a Triple X karyotype—an association not previously documented in the literature. **Conclusions**: This case highlights the potential for coexisting rare genetic conditions and underscores the critical importance of thorough and meticulous clinical evaluation.

## 1. Introduction

Congenital adrenal hyperplasia (CAH) refers to a group of autosomal recessive disorders resulting from defects in cortisol synthesis. Over 90% of CAH cases are due to 21-hydroxylase (21OHD) deficiency. In such cases, cortisol deficiency is accompanied by increased androgen production and reduced or absent aldosterone synthesis, leading to a wide spectrum of phenotypic manifestations [1]. Genetic variations in the *CYP21A2* gene, which encodes the 21-hydroxylase enzyme, include more than 300 pathogenic variants, along with macro and micro conversion events, deletions, and duplications [1,2,3].

Most CAH patients are compound heterozygotes, with the milder allele typically influencing the phenotypic expression. Phenotype–genotype correlations are well-documented [4]. The prevalence of the non-classical form (NC-CAH) is estimated to be 0.1% of the general population or higher, depending on ethnicity and race, with an incidence ranging from 1:200 to 1:1000 [5]. While the classical forms of CAH are typically diagnosed at birth, NC-CAH is usually detected later in childhood, adolescence, or adulthood [3,5]. Patients with NC-CAH exhibit relatively mild enzyme impairment and present with symptoms such as premature pubarche in children, and acne, hirsutism, menstrual irregularities, or infertility in adult females and lifelong glucocorticoid treatment is generally not required. These subtle clinical features often pose a diagnostic challenge [6].

Triple X syndrome, a relatively underdiagnosed sex chromosome abnormality, has an estimated prevalence of 1:1000 females. It is associated with heterogeneous and non-specific clinical features, including neurological, behavioral, and physical disorders [7,8]. Diagnosis is often made through prenatal screening or, less frequently, during postnatal investigations for growth or developmental delays. These may include attention-deficit hyperactivity disorder (ADHD), learning and language difficulties, anxiety, social dysfunction, autism spectrum traits, and physical abnormalities such as cardiac, renal, and infertility-related conditions, including premature primary ovarian failure (POF) [9]. Triple X syndrome arises from the presence of an extra X chromosome due to errors in meiotic division, although the specific genes involved remain unidentified [10].

Genetic abnormalities co-occurring with CAH are rarely reported, with Turner syndrome being the most commonly documented. In such cases, classic CAH is identified at birth through ambiguous genitalia, with or without salt-wasting crises in association with characteristics of Turner phenotype can aid both diagnosis [11,12]. In contrast, the NC-CAH and the milder presentation of Triple X syndrome often complicate the diagnosis, as both conditions are frequently underrecognized. Kurtoğlu et al. [13] described a unique case of CAH due to 11β-hydroxylase deficiency in a patient with a Triple X karyotype who presented at birth with ambiguous genitalia.

This study reports a novel case of a Triple X karyotype associated with NC-CAH due to 21OHD—a combination that, to the best of our knowledge, has not been previously described in the literature.

## 2. Patient Information and Clinical Findings

A 38-year-old Brazilian female patient was raised solely by her mother, with no contact with her father or paternal family. At the age of 25, she sought another endocrinological care service due to secondary amenorrhea. At that time, she was diagnosed with POF without further investigation into its etiology. Based on available information, she had been receiving estradiol valerate combined with levonorgestrel therapy, which was discontinued several months before her evaluation at our clinic.

During her initial presentation, she reported an absence of menstrual cycles along with symptoms such as hot flashes, acne, excessive skin oiliness, and hair loss. The patient maintained a regular exercise routine and followed a balanced diet. Her medical history was notable only for one orthopedic surgery, and she denied having any other comorbidities or a maternal family history of genetic diseases.

On physical examination, her vital signs were within normal limits. She had a height of 1.73 m, a BMI of 26.06, and long legs. Notable findings included thinning hair in an android pattern, acne lesions on the face and neck, and oily hair. There was no evidence of trunk hirsutism or syndromic features, such as clinodactyly or epicanthic folds. She exhibited female genitalia without clitoromegaly or hyperpigmentation.

## 3. Diagnostic Approach

### 3.1. Laboratory Assessment

The patient presented euglycemia, a normal HbA1c level, as well as normal hemogram, lipid profile, and hepatic and renal function tests. Hormonal evaluation revealed normal thyroid function, prolactin, and IGF-1 levels. However, elevated LH and FSH levels, combined with low estradiol levels, supported the diagnosis of primary ovarian insufficiency. To investigate further, a karyotype analysis was performed, revealing a 47,XXX [3]/46,XX [27] mosaicism (Figure 1).

Based on the clinical findings of hyperandrogenism, measurements of androgen levels and 17-hydroxyprogesterone (17OHP) were requested (Table 1). During the biochemical evaluation, the patient was not taking any medications. All results were confirmed with a second sample taken more than three months later. The patient demonstrated elevated morning basal levels of androgens and 17OHP (728 and 1074 ng/dL), the latter being consistent with NC-CAH due to 21-OHD. The diagnosis was further confirmed through an EV 250 μg cosyntropin stimulation test (Cosyntropin ACTH 1–24, SynacthenR, Novartis, Australia). Table 2 shows the plasma cortisol and 17OHP levels following the Synacthen stimulation, which validated the diagnosis of NC-CAH due to 21-OHD [5]. A basal 17-OHP concentration > 500 ng/mL strongly suggests NC-CAH and the definitive diagnosis requires a 17-OHP concentration ≥ 1500 ng/mL after cosyntropin-stimulation [14].

### 3.2. Molecular Analysis of the CYP21A2 Gene

Genomic DNA was obtained from peripheral blood using QIAamp DNA blood kit (QIAGEN, CA, USA). Molecular analysis was sequentially performed in three steps—allele-specific PCR (ASO-PCR), multiplex ligation-dependent probe amplification (MLPA), and direct sequencing—as previously described [15].

In summary, ASO-PCR was performed using eight pairs of primers designed to detect the common *CYP21A1P* mutations, including p.P30L, IVS2 -13 A/C>G, c.329_336del, p.I172N, CL6, p.V281L, p.R356W, and p.Q318X. The ASO-PCR analysis identified three mutations from the pseudogene: c.329_336del in exon 3, p.V281L in exon 7, and p.Q318X in exon 8, all of which were confirmed by direct sequencing.

MLPA analysis was conducted using the SALSA MLPA P050-CAH-C1-0517 kit (MRC Holland, Amsterdam, The Netherlands). Fragment analysis was performed on an ABI 3130 Genetic Analyzer (ABI PRISM/PE Biosystems, Foster City, CA, USA), and revealed a duplication of *CYP21A2* on one allele (second bar of Figure 2). The increased signal from the *CYP21A1P-3* probe (SNP del8bp-mut) confirmed the presence of the c.329_336del mutation in exon 3, detected in both *CYP21A1P* and potentially in one *CYP21A2* allele. Additionally, MLPA probes targeting *CYP21A2* regions *CYP21A2-4* (wt I173N), *CYP21A2-6* (wt V238E), *CYP21A2-6* (wt M240K), and *CYP21A2-7* (wt F308+T) showed ratios above 1.2, indicating an increase in genetic material (bars 8 to 11 of Figure 2). These findings suggest that the patient’s haplotype consists of one bimodular allele and one trimodular allele.

Two fragments of the *CYP21A2*—one from the 5′UTR to exon3 (c.329_336WT in exon 3) and another from the exon 3 (c.329_336WT in exon 3) to 3′UTR—were amplified and sequenced with internal primers using a Big Dye™ Terminator Cycle Sequencing Kit V3.1 Ready Reaction (ABI PRISM/PE Biosystems, Foster City, CA, USA). The sequences were analyzed on an ABI 3130 Genetic Analyzer and compared to the *CYP21A2* and *CYP21A1P* reference sequences (Ensembl-ENSG0000231852 and ENSG00000204338, respectively) using a Codon Code Aligner (Li-COR, Inc., Lincoln, NE, USA). The sequencing data from the exon 3 to 3′UTR fragment revealed the presence of p.V281L in exon 7 and p.Q318X in exon 8 in heterozygous (Figure 2B). Furthermore, the sequencing also confirmed that the c.329_336del mutation in exon 3 was not in combination with either p.V281L or p.Q318X, suggesting that this mutation resides on another copy of the gene.

There are two possible explanations for the patient’s haplotype based on their non-classical phenotype. One possibility is the presence of the 329_336delGAGACTAC (Δ8) and p.V281L mutations in cis with p.Q318X in trans. The other is the p.Q318X and p.V281L mutations in cis with 329_336delGAGACTAC (Δ8) in trans (Figure 2D,E).

## 4. Follow-Up and Outcomes

During the 3-year follow-up, the patient received a low-dose corticosteroid replacement therapy, which led to an improvement in symptoms related to hyperandrogenism, and a hormone replacement therapy with estradiol 1.0 mg and norethisterone acetate 0.5 mg, with good control of symptoms related to POF and a considerable improvement in quality of life.

## 5. Discussion

This report describes a patient with premature ovarian failure at the age of 25, accompanied later by mild hyperandrogenism. Molecular analysis revealed a rare association of NC-CAH due to 21OHD and a Triple X karyotype. To our knowledge, this unique association has not been previously reported in the literature.

The *CYP21A2* gene, which encodes the 21-hydroxylase enzyme, is located on chromosome 6 in the human leukocyte antigen (HLA) region, adjacent to the non-functional pseudogene *CYP21A1P.* These two genes share a 98% sequence identity. This genomic region is part of the *RCCX* module, which contains the *RP*, *C4*, *CYP21*, and *TNX* genes. Within the *RCCX* module, three pseudogenes—*CYP21A1P*, *TNXA*, and *RP2*—are situated between two C4 loci. Due to the high sequence homology and the tandem-repeat structure of the *RCCX* module, this region is prone to unequal crossover events, which can lead to large genomic rearrangements, including duplications, deletions, and fusions of the *RCCX* genes [16]. In most cases, the *RCCX* module has a bimodular structure with one *CYP21A2* gene and one pseudogene.

Our patient later presented with mild signs of virilization, raising suspicion for NC-CAH due to a *CYP21A2* mutation. Using ASO-PCR, we identified three mutations derived from the pseudogene: c.329_336del in exon 3, p.V281L in exon 7, and p.Q318X in exon 8, all of which were further confirmed by direct sequencing. The 8 bp deletion (c.329_336del) in exon 3 and the p.Q318X mutation in exon 8 are associated with the salt-wasting form, the most severe phenotype of CAH. In contrast, the p.V281L mutation in exon 7 is the primary variant responsible for NC-CAH. Since the patient’s parental DNA was unavailable, we conducted a comprehensive genetic analysis using a sequential approach combining MLPA, ASO-PCR, and direct sequencing to better provide a reliable genetic diagnosis.

MLPA analysis revealed a bimodular/trimodular haplotype, with two copies of *CYP21A1P* and three copies of *CYP21A2*, a structure found in approximately 14% of the Caucasian population [17,18]. The frameshift mutation c.329_336del in exon 3 was identified in both *CYP21A1P* copies and one of the three *CYP21A2* copies. Molecular analysis further revealed the presence of heterozygous mutation p.V281L in exon 7 and p.Q318X mutation in exon 8 in. Notably, the c.329_336del mutation in exon 3 was not found in combination with either p.V281L or p.Q318X, suggesting that the frameshift mutation resides on a separate copy of the gene. Altogether, these findings support the diagnosis of the NC-CAH form in our patient.

Classical and non-classical forms of CAH due to 21OHD have been associated with genetic syndromes, most notably Turner syndrome, with the first case documented in 1983 [12,19,20]. In many of these studies, patients were neonates, as the diagnosis of Turner syndrome mosaicism is typically made during pregnancy, often based on abnormal sonographic findings or, more recently, through amniocentesis or neonatal karyotyping/chromosome microarray. Additionally, the presence of ambiguous genitalia at birth raises the possibility of classical forms of CAH due to 21OHD or other rarer virilizing forms of CAH.

In 2004, the first case of CAH caused by 11β-hydroxylase deficiency in a 6-day-old infant with ambiguous genitalia and a Triple X karyotype was reported [13]. More recently, Liang et al. (2022) described a complex case involving a Triple X karyotype combined with the simple virilizing form of CAH due to 21OHD, as well as Graves’ disease. The patient presented with a history of primary amenorrhea, thyroid dysfunction, clitoromegaly, and hirsutism [19]. Another case of association of CAH and triple X was described by Yao et al. (2023), but the patient was a 56 year old woman with clinical features associated with the classical form of CAH due to 17α-hydroxylase deficiency, such as arterial hypertension secondary to primary hyperaldosteronism and genital dysplasia [21]. Unlike our case, none of the above cases described the presence of mosaicism, which can occur in triple X syndrome, resulting in karyotypes 47,XXX/45,X or 47,XXX/46,XX. This variability may generate different clinical presentations and the need for individualized genetic counseling [9].

Triple X syndrome is the most common sex chromosomal abnormality in women, characterized by a variable phenotype. Clinical manifestations at birth typically include low birth weight and a smaller head circumference. Toddlers with Triple X syndrome may exhibit delayed language development, while younger girls often show accelerated growth (tall stature) until puberty. Additionally, motor coordination issues and verbal and auditory processing disorders are common. In adult women aged 19–40, premature ovarian failure with amenorrhea appears to be more prevalent than in the general population [22,23]. A population-based study of women with confirmed premature ovarian failure found that X chromosome trisomy was present in 3.8% of cases [24].

Premature ovarian failure can result from a variety of etiologies, including idiopathic, autoimmune, and metabolic-infectious causes. Genetic factors account for 10–30% of cases. Among these, monogenic partial defects or loss-of-function mutations have been identified in various genes, such as those encoding ligand proteins (e.g., BMP15, GDF9, GREM1), receptors (FSHR, LHCGR), intracellular signaling molecules (DIAPH2, DNAH5, NAIP), enzymes (POLG1, ADAMTS16, ADAMTS19), DNA repair and meiosis factors (SYCE1, SMC1B, STAG3, MCM8), transcription factors (FIGLA, NOBOX, LHX8, SOHLH1), and RNA metabolism/translation proteins (FMR1, NANOS3, CPEB1) [25]. Genetic syndromes such as Turner syndrome, fragile X syndrome, and Triple X syndrome have also been associated with POF [26]. Additionally, a rare syndromic form of POF linked to adrenal insufficiency has been connected to loss-of-function mutations in the NR5A1 gene, which encodes the SF-1 transcription factor critical for ovarian development [27,28].

The main limitations of this study are related to the loss of follow-up of the patient in our service and to the unavailability of evaluation of the patient’s family members. During the follow-up period, it was not possible to perform subsequent karyotype analysis from blood samples or skin biopsy, which could provide a better assessment of mosaicism and its possible impact on the patient’s clinical condition. Furthermore, adequate genetic counseling was not performed because the patient did not show any interest.

The current study describes a 39-year-old patient whose clinical presentation began with secondary amenorrhea leading to a subsequent diagnosis of premature ovarian failure associated with X trisomy, a heterogeneous condition that can result in amenorrhea and infertility in women [29]. Additionally, the patient exhibited hyperandrogenism, which led to the diagnosis of NC-CAH due to a *CYP21A2* mutation. While this association could be a coincidental finding, it is noteworthy for demonstrating a previously undocumented genotype–phenotype link.

## 6. Conclusions

This case underscores the importance of comprehensive clinical evaluation, including patient history, physical examination, and genetic studies, in managing complex medical cases. In our patient, targeted treatment for both conditions is expected to improve outcomes. This report not only enhances understanding of the association between Triple X syndrome and CAH but also emphasizes the significance of exploring novel genetic findings and clinical correlations in rare disorders.

## Figures and Tables

**Figure 1 genes-16-00354-f001:**
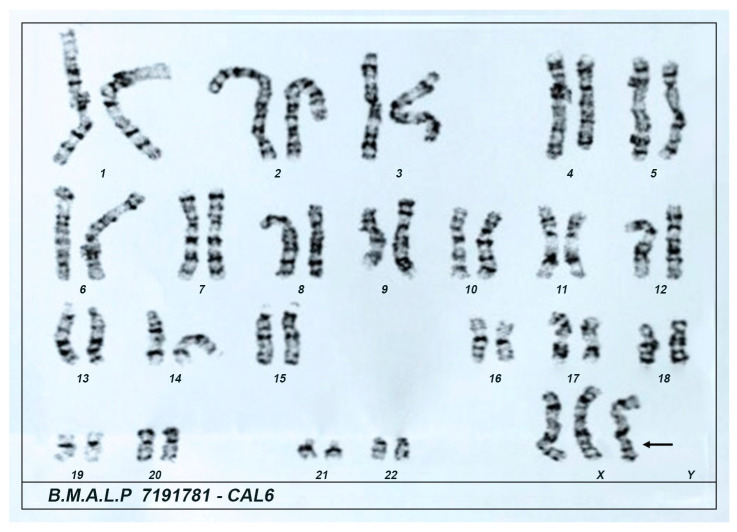
Patient’s karyotype at 39 years of age showing the presence of the triple X as the single chromosomal aberration (47,XXX [3]/46,XX [27]). Material collected: blood. Method: G-banding. Analysis of 30 cells with a resolution of 400 bands. Karyotype 47,XXX was found in 10% of the cells.

**Figure 2 genes-16-00354-f002:**
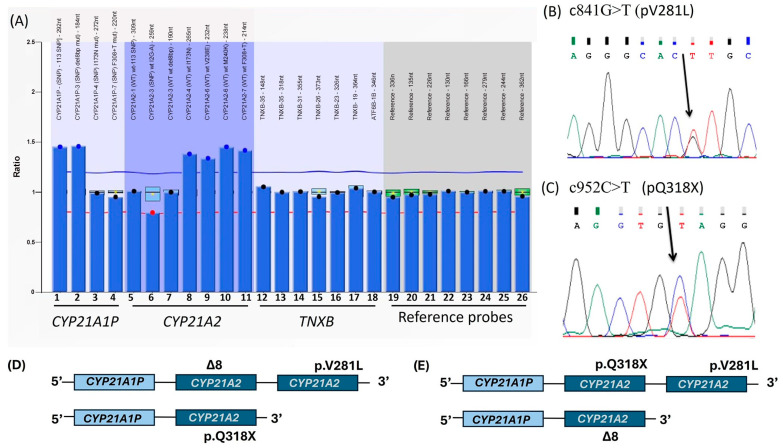
Multiplex ligation-dependent probe amplification (MLPA) of *CYP21A2* of the studied patient. (**A**) Each column represents the integrated and normalized peak areas using Coffalyser from the electropherogram. Values ranging between 0.8 and 1.2 indicate the presence of two copies of the gene, values below 0.8 suggest a single gene copy, and values above 1.2 indicate more than two copies. The graph highlights a duplication of *CYP21A1P* visualized in probes 1 and 2 as well as an increase in genetic material (bars 8 to 11) associated with *CYP21A2* revealing two copies of *CYP21A1P* and three copies of *CYP21A2* in a bimodular/trimodular haplotypes carrying the 329_336delGAGACTAC mutation. (**B**) Direct sequencing of the *CYP21A2* fragment spanning exon 3 (3EX) to the 3′ untranslated region (3′UTR) revealing a heterozygous c.841G>T substitution (p.V281L) in exon 7 and (**C**) a heterozygous c.952C>T substitution (p.Q318X) in exon 8. Schematic representation of the molecular data of the studied patient. (**D**) The 329_336delGAGACTAC (Δ8) and p.V281L mutations in *cis* with p.Q318X in *trans*. (**E**) The p.Q318X and p.V281L mutations in *cis* with 329_336delGAGACTAC (Δ8) in *trans*.

**Table 1 genes-16-00354-t001:** Hormonal parameters in the reported patient.

Hormones	First Sample	Confirmatory Sample	Reference Values
TSH	0.59 μIU/mL	0.85 μIU/mL	0.4–4.5 μIU/mL
FT4	1.2 μIU/mL	1.4 μIU/mL	0.7–1.8 μIU/mL
IGF-1	NA	214 ng⁄mL	5–40 years 109.0–284.0 ng/mL
Prolactin	11 ng/mL	19 ng/mL	Non-pregnancy < 30 ng/mL
FSH	90 IU/L	105 IU/L	Post-menopause > 30 IU/L
LH	50 IU/L	69 IU/L	Post-menopause 5.2 to 62.9 IU/L
17-OHP	728 ng/dL	1074 ng/dL	Follicular phase: up to 110 ng/dL Luteal phase: 86 to 400 ng/dL
Testosterone	52 ng/dL	84 ng/dL	Women > 21 years: 12.0–59.5 ng/dL
SDHEA	NE	605 µg/dL	35–44 years: 74.8–410 µg/dL

TSH—thyroid stimulating hormone; FT4—thyroxine free; IGF-1—insulin-like growth factor 1; FSH—follicle stimulating hormone; LH—luteinizing hormone; 17-OHP—17-hydroxyprogesterone; SDHEA—dehydroepiandrosterone sulfate; NE—not evaluated.

**Table 2 genes-16-00354-t002:** Plasma cortisol and 17-hydroxyprogesterone levels in basal condition and post-ACTH test in the reported patient.

Time	Cortisol	17-OH Progesterone
Basal 0′	10.3 NR: 5.0–25.0 µg/dL	658 NR: FP 20–130 ng/dL NR: LP 100–450 ng/dL
After 30′	14.7 NR: >18 µg/dL	1699 1500–6000 ng/dL confirm NC-CAH
After 60′	17.1 NR: >18 µg/dL	5808 >1500–6000 ng/dL confirm NC-CAH

Plasma cortisol and 17-hydroxyprogesterone were measured by electrochemiluminescence and enzyme immunoassay methods, respectively. NR—normal range; FP—follicular phase; LP—luteal Phase. NC-CAH: nonclassical congenital adrenal hyperplasia.

## Data Availability

The original contributions presented in this study are included in the article. Further inquiries can be directed to the corresponding author.

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
