# Peer review of "Unique Case Report: A Rare Association of 21-Hydroxylase Deficiency with Triple X Karyotype"

_genes, 2025, doi:10.3390/genes16030354_

Round 1
Reviewer 1 Report
Comments and Suggestions for Authors
This case report adds to the literature by describing a rare patient diagnosed clinically and then molecularly with premature ovarian failure, hyperandrogenism, non-classic congenital adrenal hyperplasia (NC-CAH) syndrome and CYP21A2 mutations. Its major interest lies in the association and clinical overlap with this patient's concomitant Triple-X syndrome.
It is clearly written and the molecular exploration and explanations (particularly about the CYP21A1P pseudogenes, but also the endocrinological exploration) was thorough and convincing. I have some minor suggestions that could allow a broader readership to better understand why this case is of special interest.
- The paper mentions that the patient is mosaic for the third X chromosome from karyotyping blood cells. What is the percentage of mosaicism? Was there a skin or any other biopsy, in which it could be determined in an ectodermal derivative as well?
- Given that NC-CAH is more frequent within certain ethnic groups with high consanguinity, was the maternal family further investigated?
- There appears to be a similar case published recently that the authors should also discuss: https://pubmed.ncbi.nlm.nih.gov/36820210/
- Does the patient intend to have children? What was the genetic counseling to her? What was the targeted treatment recommended (cf line 243)?
- In the Discussion, the second and third paragraphs could be clarified with a simple diagram added on the bottom part as Figure 2D. CYP21A1P is non-functional, but in lines 179-180 it states that “three mutations derived from the pseudogene” were found, and at lines 188-189, that there were “two copies of CYP21A1P and three copies of CYP21A2”. These seem to imply that CYP21A1P is transcribed in some manner and can confuse things for the reader. A consistent model of the mutations in cis/trans and the proposed non-disjunction event could help.
Typographical issues:
- Genes and their transcripts should be italicized throughout as they are in figure 2.
- Line 105, add “with” to “consistent… NC-CAH”
- Line 192-193, the presence of heterozygous mutations p.V281L in exon 7 and p.Q318X in exon 8.
Author Response
Thank you very much for taking the time to review this manuscript. Please find the detailed responses below and the corresponding revisions in the re-submitted files
- We added the following sentence to line 99, in Figure 1 legend "Karyotype 47, XXX was found in 10% of the cells."
Unfortunately, we were not able to perform a skin or any other biopsy during the patient’s follow-up.
2. The DNA from parents was not available, Thus, in page 6, line 198-201, we clarify this issue with the sentence “Since the patient's parental DNA was unavailable, we conducted a comprehensive genetic analysis using a sequential approach combining MLPA, ASO-PCR, and direct sequencing to better provide a reliable genetic diagnosis.”
3. We added the following commentary to Discussion, on page 7, lines 224-227 : “Another case of association of CAH and triple X was described by Yao et al (2023), but the patient was a 56 year old woman with clinical features associated to the classical form of CAH due to 17α-hydroxylase deficiency, such as arterial hypertension second-ary to primary hyperaldosteronism and genital dysplasia [21].”
- The patient did not intend to have children, so genetic counseling, although offered, was not in her interest.
- Regarding non-disjunction event, due to the presence of multiple copies of CYP21A1P and CYP21A2 and the fact that these two genes share a 98% sequence identity, there is a high possibility of unequal distribution of chromosomes leading to the observed genetic variations. As suggested, to clarify the issues on lines 179-180 and 188-189, we added a diagram showing the possible model illustrating the mutations in cis/trans. In addition, we added in the text, page 5, lines 167-170. the following sentence: There are two possible explanations for the patient's haplotype based on their non-classical phenotype. One possibility is the presence of the 329_336delGAGACTAC (Δ8) and p.V281L mutations in cis with p.Q318X in trans. The other is the p.Q318X and p.V281L mutations in cis with 329_336delGAGACTAC (Δ8) in trans (Figure 2D and 2E). In addition, we modified the legend as follows: Figure 2. Multiplex Ligation Dependent Probe Amplification (MLPA) of CYP21A2 of the studied patient. (A) Each column represents the integrated and normalized peak areas using Coffalyser from the electropherogram. Values ranging between 0.8 and 1.2 indicate the presence of two copies of the gene, values below 0.8 suggest a single gene copy, and values above 1.2 indicate more than two copies. The graph highlights a duplication of CYP21A1P visualized in probes 1 and 2 as well as an increase in genetic material (bars 8 to 11) associated with CYP21A2 revealing two copies of CYP21A1P and three copies of CYP21A2 in a bimodular/trimodular haplotypes carrying the 329_336delGAGACTAC mutation. (B) Direct sequencing of the CYP21A2 fragment spanning exon 3 (3EX) to the 3' untranslated region (3’UTR) revealing a heterozygous c.841G>T substitution (p.V281L) in exon 7 and (C) a heterozygous c.952C>T substitution (p.Q318X) in exon 8. Schematic representation of the molecular data of the studied patient. (D) The 329_336delGAGACTAC (Δ8) and p.V281L mutations in cis with p.Q318X in trans or (E) The p.Q318X and p.V281L mutations in cis with 329_336delGAGACTAC (Δ8) in trans.
Reviewer 2 Report
Comments and Suggestions for Authors
The paper under review is an interesting case report that underlines the necessity of detailed genetic evaluation the patients so as not to miss the occurrence of several disorders in the same patient. In my opinion, the manuscript is appropriate for publication after minor revision. Below you can find few detailed remarks.
There is an inconsistence in reported patient’s age in Abstract (38 years), in the main text (45 years – line 72, with data suggesting follow-up from the age of 25 years – line 73) and in Discussion (25 years) – please unify and explain this difference.
If the authors really have a long-term follow-up of the patient, it would be interesting to provide a short information about treatment following the diagnosis.
Last line in Table 1 and description below – there is NA in the Table, while the abbreviation NE explained – please correct.
Abbreviation POF is introduced in line 57 and – for the second time – in line 74, and - once more – in line 218, please correct.
Author Response
Thank you very much for taking the time to review this manuscript. Please find the detailed responses below and the corresponding revisions/corrections in the re-submitted files.
-
we corrected the mistake. The actual age of the patient is 38 years old. We also corrected line 73 to clearify that at the age of 25 she was first evaluated by other endocrinology service, with no further investigation.
2- We also added to the text the session “4. Follow-up and outcomes”, on page 6, lines 171-176.
All other minor mistakes were corrected in the text.